# Performance Evaluation of Thermal Bridge Reduction Method for Balcony in Apartment Buildings

Xinwen Zhang [ID], Gun-Joo Jung [ID] and Kyu-Nam Rhee *

Department of Architectural Engineering, Pukyong National University, Busan 48513, Korea; moon@pknu.ac.kr (X.Z.); envjung@pknu.ac.kr (G.-J.J.)
* Correspondence: knrhee@pknu.ac.kr; Tel.: +82-51-629-6090

**Abstract:** Most apartment buildings in South Korea use internal insulation systems to reduce building energy demand. However, thermal bridges such as balcony slabs in apartment buildings still lead to significant heat loss in winter, because the internal insulation system is not continuous in the balcony slab structure, and floor heating systems are commonly used in residential buildings. Therefore, this study investigates two types of thermal break elements, namely thermal break (TB) and thermal break-fiber glass reinforced polymer (TB-GFRP), to improve the thermal resistance of a balcony thermal bridge. To understand the effects of balcony thermal bridges with and without thermal break elements, the linear thermal transmittances of different balcony thermal bridges were analyzed using Physibel simulations. Then, the heating demand of a model apartment under varying thermal bridge conditions was evaluated using TRNSYS simulations. To understand the effect of insulation systems on heat loss through a balcony thermal bridge, apartments with internal and external insulation systems were studied. Whether the apartment was heating was also considered in the thermal transmittance analysis. Thus, the linear thermal transmittance of the thermal bridges with thermal break elements was reduced by more than 60%, and the heating energy demands were reduced by more than 8%.

**Keywords:** balcony slab thermal bridge; floor heating; linear thermal transmittance; thermal break; insulation system; heating energy demand



## 1. Introduction

Persistent environmental problems, such as the lack of fossil fuels and global warming, are still major challenges for humans. Therefore, policies for energy saving and emission reduction have been adopted worldwide. The EU has a target to improve energy efficiency to 32.5% by 2030 to reduce energy consumption [1]. South Korea plans to reduce $CO_2$ emissions by 37% by 2030 compared with business-as-usual (BAU) [2]. Thus, reducing energy consumption is necessary to improve environmental problems. Among the total energy consumption, building energy consumption contributes over 30% [3], and 70.4% of the building energy is consumed by apartment buildings [4]. Apartment buildings have less internal load compared to other buildings, hence they are envelope-load-dominated buildings. Therefore, heating load accounts for a large part of the total residential load, and more energy is consumed for heating. To reduce apartment building energy consumption, the performance of building equipment systems is improved, and innovative building components are used. In addition, a part of the heating load comes from the heat loss through thermal bridges of the building envelope, which in winter can be up to 30% of the total apartment building energy consumption [5,6]. Therefore, apartment buildings have a large potential for reducing energy consumption by improving the effects of thermal bridges. Therefore, increased attention is being paid to construction details where thermal bridges are present.

The thermal bridge occurs from the high thermal transmittance material in the insulated structures of buildings, which leads to a significant increase in the heat flow through

the wall [7,8]. As a result, thermal bridges increase the additional heat loss in winter and heat gain in summer [9,10]. Furthermore, the heat loss caused by the thermal bridge in winter leads to a decrease in the temperature of the inner surface and an increase in condensation [11]. In apartment buildings, windows, doors, and junctions are weak areas that can form thermal bridges [12–14]. The balcony slab is the second-largest component of the thermal bridge in buildings, except for windows and doors. Ge et al. [15] stated, owing to the windows and opaque walls installed in the balcony, the space heating energy consumption of a typical high-rise apartment building may increase up to 11% and the internal floor surface temperature tends to be reduced, which can cause negative effect on the indoor thermal comfort. Kotti et al. [16] estimated the impact of thermal bridges, including windows and balcony slabs, on the overall annual load of 13% in an apartment building. Ge and Baba [17] evaluated the effect of thermal bridges on the energy performance of a low-rise apartment building, and found that when considering typical junctions and balcony slabs, the annual heating load was increased by 30% in the cold climate. In addition to the windows and opaque walls of the balcony, the reinforced concrete slab is also the main part of the thermal bridge. In apartment buildings of South Korea, it is common to enclose most balconies with external windows and to install radiant floor heating systems. As balconies are not included in air-conditioned zones or heating zones, the balcony slabs penetrate the walls, separating the heating and non-heating zones. Under this condition, the heat loss through the balcony slabs can significantly increase [18], as hot water pipes for radiant floor heating system can increase the temperature difference between the heating and non-heating zones [19].

An effective solution to reduce the heat loss caused by thermal bridges is to improve the thermal resistance of the places where thermal bridges are formed [20]. To prevent heat loss due to thermal bridges, the first step is to apply more insulation to building envelopes [21,22]. Hallik et al. [23] explained that a well-insulated building structure is crucial for zero energy building, however, the thermal bridge can deteriorate the overall thermal performance of the buildings. Building insulation systems generally can be divided into external and internal insulation systems. Mostly, the external insulation system performs better than the internal insulation system. El saied et al. [24] evaluated the heat loss due to thermal bridges by adding variety insulation systems (internal and external), and the results showed that the external insulation can reduce the thermal bridge effect by 53–63%. This is because the external insulation system uses exterior continuous insulation in walls, which reduces the risk of thermal bridge formation [25]. Li et al. [26] added a local insulation layer of the L-shaped and T-shaped structure of a concrete wall to mitigate the influence of the thermal bridge. Ibrahim et al. [27] added an insulating coating to the window offset and evaluated the cooling and heating loads of a typical French house building. As a result, 1 cm and 2 cm of coating reduce by 24–50% energy load from the window offset. To avoid heat loss by balcony thermal bridges, Karabulut et al. [28] analyzed the effect on the insulating surfaces of an intermediate floor beam wall and an extended balcony board. This study illustrated the effects of the thickness and type (external and internal) of the insulation system on the thermal performance of balconies with different insulation materials. The results showed that an external insulation system is more effective than an internal one. Murad et al. [29] insulated a concrete curb on a concrete balcony slab, and found the interior surface temperature of the slab with the insulated concrete curb increased. Baba and He [30] showed that a balcony slab with external insulation can reduce the heating load of a high-rise apartment building by 8.8–25.7%.

However, most apartment buildings in South Korea are constructed with internal insulation systems, and the wall insulation is usually not continuous at the wall–slab junction even though external insulation systems are applied; therefore, the thermal insulation effect of the balcony slab thermal bridge is often limited [31]. Meanwhile, with the development of the construction industry and building materials' research and development, building insulation standards have increased significantly, and therefore the heat loss will be more concentrated at the thermal bridges. For this reason, different methods for reducing heat

loss by balcony slab thermal bridges have been investigated in many studies. In addition to improving the insulation system, installing thermal break (TB) in the balcony thermal bridge is an effective way to prevent heat loss.

A TB is an element of low thermal conductivity placed in thermal bridges to avoid excessive heat loss between the indoor and outdoor environments. It is designed so that it can mitigate the direct heat flow through the balcony slabs while providing sufficient strength to resist the shear force at the wall–slab junctions. In the study conducted by Santos et al. [32], it was found that the inner or outer TB strips in the light steel-framed building components could effectively improve the thermal resistance, and double TB strips could lead to more enhanced thermal performance. However, a balcony slab with a conventional TB still has a large heat transfer because of the stainless-steel bars in the concrete slab. In general, a TB product for balcony slabs consists mainly of an insulation block and stainless-steel bars. To prevent heat transfer from the thermal bridge more effectively, a noncorrosive material for reinforcement bars can be an alternative to TB components. Thus, a glass fiber-reinforced polymer (GFRP), which has a relatively low cost to be implemented, has been used as an alternative material for producing reinforcing bars [33,34]. Wakili et al. [35] introduced a TB with GFRP (TB-GFRP) for the concrete balcony slab, and analyzed the thermal performance of the proposed balcony slab. This study showed that the TB-GFRP element reduced the additional heat loss compared to the balcony slab without TB-GFPR. Goulouti et al. [36] also showed that a building using TB-GFRP can reduce the heating load by 41%. Even if GFRP can effectively reduce the heat transmittance, the application of GFRP is still quite new. In addition, the performance of TB-GFRP was not evaluated for the residential building with radiant floor heating systems, wherein the heat transfer at wall–slab junctions would be higher than that of the buildings with air heating systems.

The objective of this study was to evaluate different methods for improving the balcony slab thermal bridge. To do this, TB and TB-GFRP elements were selected as TBs, and effects of these elements on balcony thermal bridges were analyzed with the heat transfer simulations. Because internal and external insulation systems applied in apartments also have different effects on the thermal bridge, the analytical conditions were divided into two types: apartments with internal insulation systems and external insulation systems. In addition, to understand the effect of the radiant floor heating system on the balcony thermal bridge, the heating conditions in this study were also divided into an air heating system and a radiant floor heating system. The impact on the thermal bridge reduction was evaluated by analyzing the linear thermal transmittances and heating energy demands. In general, the linear thermal transmittances of thermal bridges were analyzed using a 2D heat transfer models [37]. Considering that the rebar in the TB is installed along the thermal bridges, 3D heat transfer simulations were conducted using Physibel–TRISCO. In addition, heating energy demands were analyzed with TRNSYS simulations, wherein thermal bridge effects were represented with the linear thermal transmittance obtained from Physibel–TRISCO simulations. In order to compare the TB alternatives with the existing thermal bridges, the balcony slab without TB was designated as reference cases for the evaluation of linear thermal transmittance and heating energy demands.

The remainder of this paper is organized as follows: Section 2 introduces the analysis cases and evaluation methods of the effects of TBs on the balcony slab thermal bridge and the heating energy demand; Section 3 describes the analysis results of the linear thermal transmittance and heating energy demand for different cases; Section 4 discusses the implications and limitations of this study; and Section 5 presents the conclusions of this study.

## 2. Methodology

### 2.1. Evaluation Cases

To understand the balcony thermal bridges, a typical balcony slab attached to an exterior wall was modeled as the object of the analysis. There are three analytical conditions: a wall with external and internal insulation systems, an interior with an air heating system

or radiant floor heating system, and a balcony slab with and without a TB. The original concrete balcony slab without any TBs was used as the reference case. As shown in Figure 1a, the thermal conductivity of steel reinforcing bar used in the balcony slab was 50 W/mK [38]. The analysis data of the reference cases were used to compare the results of the balcony thermal bridge with TBs.

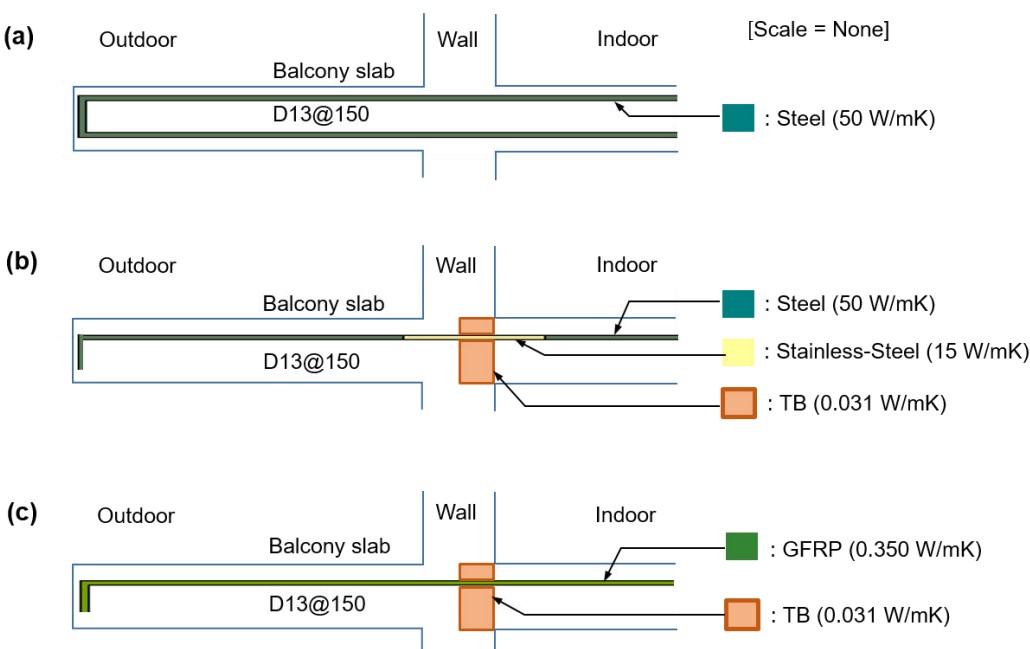

**Figure 1.** Diagram of (**a**) Reference (steel), (**b**) TB element and (**c**) TB-GFRP element.

Two design options for thermal breaks were evaluated. To improve the effect of the balcony thermal bridge, the TB element was chosen first. As shown in Figure 1b, the TB element is mainly composed of an insulation block (insulation material composed with polystyrene hard foam) and stainless bars. The conductivities of the insulation block and stainless bars of the TB element are 0.031 W/mK [38] and 15 W/mK [38], respectively. When the TB element is installed in a balcony slab, the continuous steel reinforcing bars with high thermal conductivity are replaced by the stainless-steel bars in the TB element. Therefore, the TB element is effectively used for thermal break because the excessive heat flow through the reinforcing bars can be mitigated. However, the balcony slab with the TB element still has more potential to reduce heat loss, because the remaining stainless-steel reinforcing bars in the balcony slab can become a heat flow path.

Therefore, a TB consisting of a GFRP rebar (TB-GFRP) was suggested to minimize heat flow through reinforcing bars, as described in Figure 1c. As mentioned in Section 1, GFRP rebar is a kind of non-conductive reinforcing bar, which has thermal conductivity of 0.35 W/mK [39], it has a low cost-to performance ratio, and is noncorrosive in nature [33]. Furthermore, the high tensile strength of the GFRP rebar makes it suitable for use as a steel reinforcing replacement. In the TB-GFRP element, the GFRP rebar replaced all the steel reinforcing bars of the concrete slab. To explain the thermal resistance of the GFRP rebar, the conductivity of the insulation block in the TB-GFRP element was set to the same value as that of the TB element. Table 1 shows the parameter comparison between the steel and GFRP, and Table 2 shows the material properties of the external wall and balcony slab. The parameter values were determined based on the data from the actual residential buildings.

**Table 1.** Parameters of steel and GFRP for reinforcing bars.

| Parameters | Steel | GFRP |
|---|---|---|
| Tensile strength | 450–690 MPa [40,41] | 170–270 MPa [40] |
| Density | 7830–7850 kg/m$^3$ [41] | 1870 kg/m$^3$ [42] |
| Thermal expansion coefficient | $12 \times 10^{-6}/°C$ [41] | $20 \times 10^{-6}/°C$ [40] |
| Thermal conductivity | 50 W/mK [38] | 0.350 W/mK [39] |

**Table 2.** Material properties of the external wall and balcony slab.

| Construction | Material | Thickness | Thermal Conductivity |
|---|---|---|---|
| Wall | Concrete | 210 mm | 1.35 W/mK |
| | Wall insulation | 80 mm | 0.030 W/mK |
| | Finishing material | 18 mm | 0.14 W/mK |
| Slab | Anti-condensation insulation | 20 mm | 0.030 W/mK |
| | Concrete | 210 mm | 1.35 W/mK |
| | Autoclaved light weight concrete | 50 mm | 0.19 W/mK |
| | Screed (Mortar) | 40 mm | 1.0 W/mK |
| | TB (Insulation block) | 210 mm | 0.035 W/mK |
| | Reinforcement bar (Steel) | 13 mm | 50 W/mK |
| | Reinforcement bar (Stainless-steel) | 13 mm | 15 W/mK |
| | Reinforcement bar (GFRP) | 13 mm | 0.350 W/mK |

In addition, various insulation systems applied in buildings can affect the heat transmittance of thermal bridges. Internal insulation systems have been commonly applied to apartment buildings in Korea because of easy installation and low construction cost. However, the application of external insulation systems is steadily increasing to achieve energy saving target for building sectors [43]. Therefore, the analysis cases were divided into balcony thermal bridge models with internal and external insulation systems (Figure 2 and Table 3).

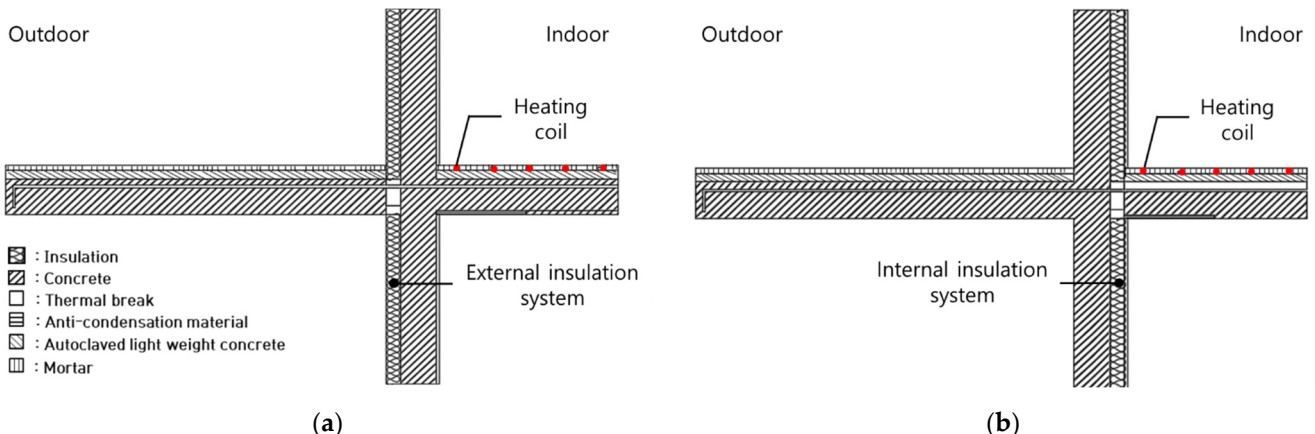

**Figure 2.** Section view of the balcony slab: (**a**) External insulation system; (**b**) Internal insulation system.

**Table 3.** Simulation cases.

| Case | | Reinforcing Bar | Thermal Break | Insulation Method | Heating System |
|---|---|---|---|---|---|
| Case 1 | (Reference) | Steel | No [a] | External | Air heating |
| Case 2 | | Steel + Stainless Steel | Yes | External | Air heating |
| Case 3 | | GFRP | Yes | External | Air heating |
| Case 4 | (Reference) | Steel | No | External | Radiant floor heating |
| Case 5 | | Steel + Stainless Steel | Yes | External | Radiant floor heating |
| Case 6 | | GFRP | Yes | External | Radiant floor heating |
| Case 7 | (Reference) | Steel | No | Internal | Air heating |
| Case 8 | | Steel + Stainless Steel | Yes | Internal | Air heating |
| Case 9 | | GFRP | Yes | Internal | Air heating |
| Case 10 | (Reference) | Steel | No | Internal | Radiant floor heating |
| Case 11 | | Steel + Stainless Steel | Yes | Internal | Radiant floor heating |
| Case 12 | | GFRP | Yes | Internal | Radiant floor heating |

[a] No: the balcony slab without thermal break; Yes: the balcony slab with thermal break.

The standards of evaluating the thermal bridge are usually based on the assumption that the indoor condition is maintained by air heating systems [19]. However, most of residential buildings in Korea are equipped with radiant floor heating systems [44], wherein the indoor condition is maintained by heated floor structure. Thus, simulation models were set with and without a radiant floor heating system in order to investigate the effect of the radiant floor heating system on the balcony thermal bridge. The model without the floor heating system was assumed to be heated with an air heating system. Under different simulation conditions, the slab balcony with steel reinforcing bars was used as the reference case. Details of the analysis cases are listed in Table 3.

### 2.2. Simulation Modeling

Thermal bridges at the wall–balcony slab junction can be considered as linear thermal bridges, which can be evaluated using linear thermal transmittance defined in ISO 10211 [45]. Linear thermal transmittance is defined as the heat transfer per unit length when the temperature difference between the indoor and outdoor environments is 1 K under steady conditions. It can be calculated by dividing the heat flow from the thermal bridge by the length of the thermal bridge and the difference between the indoor and outdoor temperatures, as formulated by Equation (1).

$$\psi = L_{2D} - \sum_{j=1}^{N_j} U_j l_j \tag{1}$$

where, $\psi$ is the linear thermal transmittance Psi of the linear thermal bridge separating the two environments being considered (W/mK), $L_{2D}$ is the thermal coupling coefficient obtained from a 2-D calculation of the component separating the two environments being considered (W/mK) $U_j$ is the thermal transmittance of the 1-D component j separating the

two environments being considered (W/m²K), $l_j$ is the length within the 2-D geometrical model over which the value $U_j$ applies (m), and Nj is the number of 1-D components.

Although $U_j$ and $l_j$ can be estimated through simple calculation, the thermal coupling coefficient $L_{2D}$ needs to be calculated with a numerical method because two-dimensional heat transfer should be analyzed. In this study, Physibel–TRISCO simulation was used to obtain the thermal coupling coefficient, which is calculated by dividing the total heat flow through the entire structure by temperature difference (indoor and outdoor temperature difference). With the Physibel simulation result, the linear thermal transmittance $\psi$ was calculated using Equation (1).

Figure 3 shows the material properties and boundary conditions used in the Physibel simulations. The same structure layer and materials were applied to all simulation cases, except for the case of TBs. Internal and external insulation systems were also analyzed assuming the same material and thickness. For boundary conditions, the indoor and outdoor temperatures were set to 20 °C and 0 °C, respectively. Hot water temperature and flow rate for radiant floor heating system was assumed as 70 °C and 0.014 kg/s, which corresponds to the internal heat transfer coefficient of 806 W/m²K.

| Col. | | Type | CEN-rule | Name | Pat. | E [W/mK] | A [-] | θ [°C] | h [W/m²K] | q [W/m²] |
|---|---|---|---|---|---|---|---|---|---|---|
| **1** | | BC_SIMPL | NIHIL | | | | | 0.0 | 0.00 | 0 |
| 5 | | MATERIAL | | TB block (Insulation) | | 0.031 | | | | |
| 6 | | BC_SIMPL | NIHIL | Hot water for radiant floor heating | | | | 70.0 | 806.01 | 0 |
| 13 | | MATERIAL | | Steel rebar | | 50.000 | | | | |
| 18 | | MATERIAL | | EPDM in TB block | | 0.250 | | | | |
| 57 | | MATERIAL | | Screed (Mortar) | | 1.000 | | | | |
| 88 | | MATERIAL | | Concrete | | 1.350 | | | | |
| 89 | | MATERIAL | | Stainless steel | | 15.000 | | | | |
| 133 | | MATERIAL | | Insulation | | 0.030 | | | | |
| 144 | | MATERIAL | | Interior finish | | 0.140 | | | | |
| 156 | | MATERIAL | | Autoclave lightweight concrete | | 0.190 | | | | |
| 170 | | BC_SIMPL | HE | Outdoor | | | | 0.0 | 25.00 | 0 |
| 174 | | BC_SIMPL | HI_NORML | Indoor | | | | 20.0 | 7.70 | 0 |

**Figure 3.** Material properties and boundary conditions in Physibel simulation.

The linear thermal transmittance was adopted to evaluate the heating energy demands as well as the impact of the TBs on the thermal bridge reduction. Heating energy demands of apartments with different balcony slab structures were analyzed using TRNSYS simulation, wherein the linear heat transmittances of the thermal bridges were inputted as parameter of external walls. To evaluate the heating energy demand, a simulation model for a typical apartment house in Korea was developed as shown in Figure 4. The investigated apartment was composed of four heated zones: a living room and three bedrooms, which is one of the most common configurations in Korea. The location of the building was assumed as Incheon, which is located in the central region of South Korea. It was assumed that the main orientation of the building is the South and the balcony is attached to the South of the building.

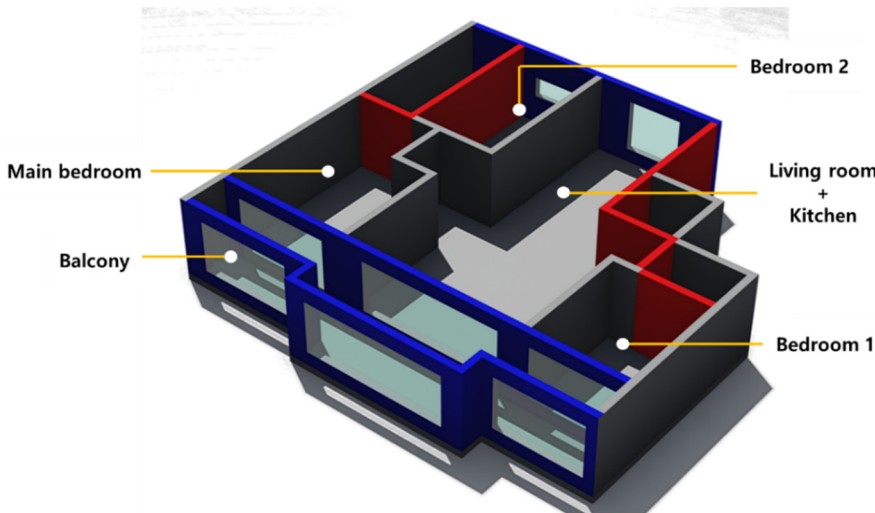

**Figure 4.** 3D view of the investigated apartment house.

TRNSYS simulation models were developed in accordance with the above-mentioned Physibel simulation models, including the structure layers, external and internal thermal insulation systems, and two types of TB elements. Therefore, the heating energy demand of the apartment was also analyzed with radiant floor heating and air heating system. The simulation of heating energy demand was performed for a heating period of one year, with hourly time step. The heating period was defined based on the input weather data. EnergyPlus weather (EPW) data of Incheon [46] was applied to utilized for the annual simulation. In order to maintain the same boundary conditions with Physibel simulation, the indoor heating temperature was set to 20 °C. The occupancy schedule was set from 6 p.m. to 9 a.m. of the next day. In this study, the peak heating load and seasonal heating load (heating energy demand) were analyzed to illustrate the energy performance of different types of balcony slab structures. Table 4 shows the summarized TRNSYS simulation conditions.

**Table 4.** Input parameters of TRNSYS simulation.

| Category | Item | Description |
|---|---|---|
| General | Location | Incheon, South Korea |
| | Weather data | EPW data for Incheon |
| | Floor area | 110 m$^2$ |
| | Orientation | South |
| Construction | External Wall | U-value = 0.322 W/m$^2$K |
| | Floor | U-value = 0.426 W/m$^2$K |
| | Window | U-value = 2.7 W/m$^2$K, SHGC = 0.6 |
| | Window-to-Wall ratio | 35% |
| | Infiltration rate | 0.5 ACH (Heated zone), 2.0 ACH (None-heated zone) |
| Heating condition | Set-point temperature | 20 °C [47] |
| | Set-back temperature | 18 °C |
| | Operation schedule | ON: 18:00 to 24:00, 00:00 to 09:00/OFF: 09:00 to 18:00 |

## 3. Simulation Results

### 3.1. Linear Thermal Transmittance

#### 3.1.1. Air Heating Condition

Table 5 shows the simulation results of heat transfer at the wall–slab junction under air heating condition. It is shown that the reference cases (Cases 1 for external insulation

and Case 4 for internal insulation) resulted in relatively large heat flow through the balcony slab, and slightly lower surface temperature at the wall–slab junction. In Cases 2, 3, 5, and 6 wherein the thermal break was applied, heat flow through the balcony slab was significantly reduced regardless of the insulation method.

**Table 5.** Simulation results of heat transfer under air heating condition.

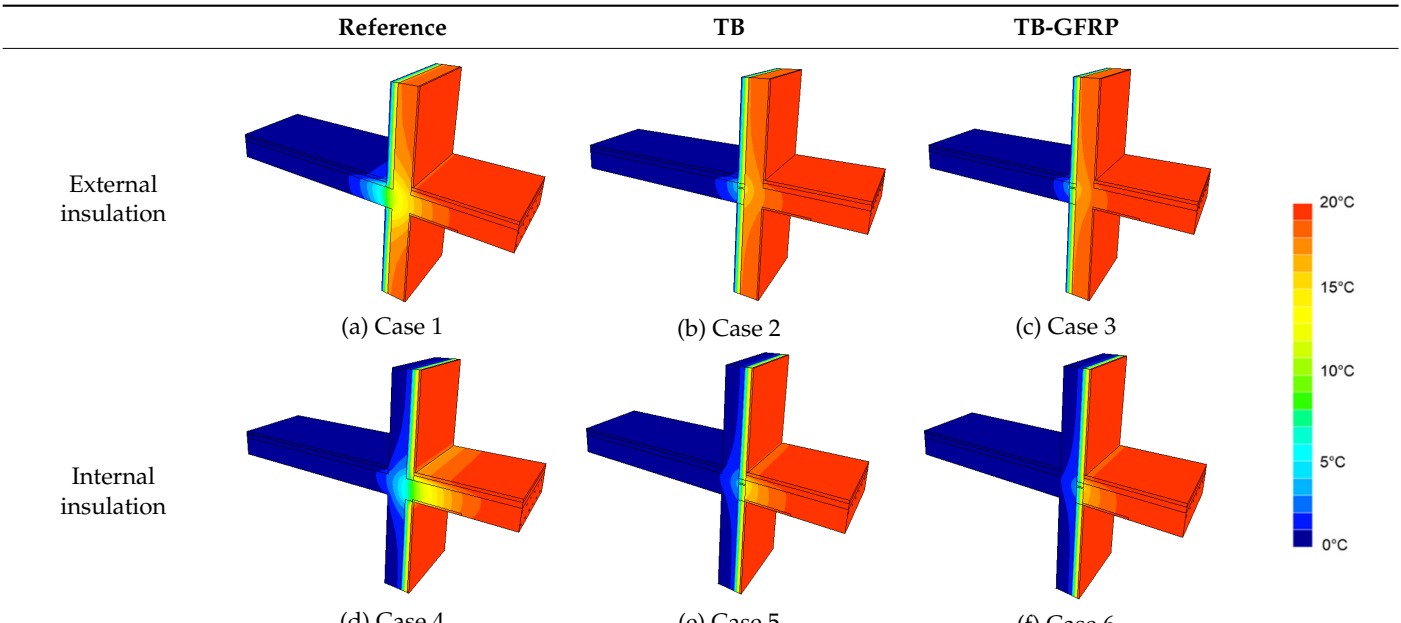

| Reference | TB | TB-GFRP |
| --- | --- | --- |
| External insulation (a) Case 1 | (b) Case 2 | (c) Case 3 |
| Internal insulation (d) Case 4 | (e) Case 5 | (f) Case 6 |

Figure 5a shows the heat flow through the construction including the thermal bridge. With an external insulation system, the total heat transfer of the reference case (Case 1), TB (Case 2), and TB-GFRP (Case 3) was 25.41 W, 17.48 W, and 16.32 W, respectively. Compared to the reference case, the reduction rate of the heat loss at the thermal bridge was 31.2 % for TB and 35.8% for TB-GFRP case. It was found that the TB-GFRP can reduce the heat loss from the balcony thermal bridge by 4.6% compared to TB.

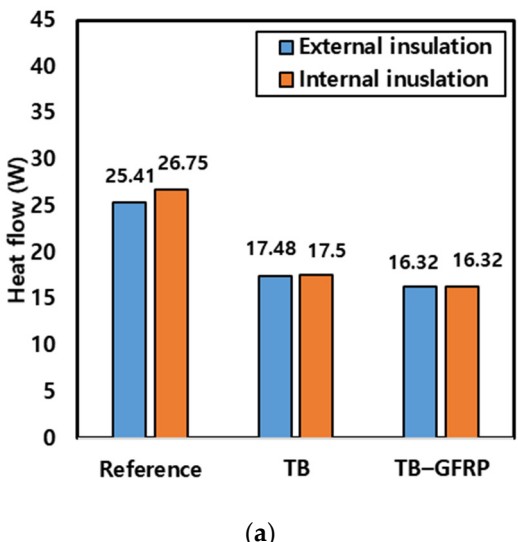

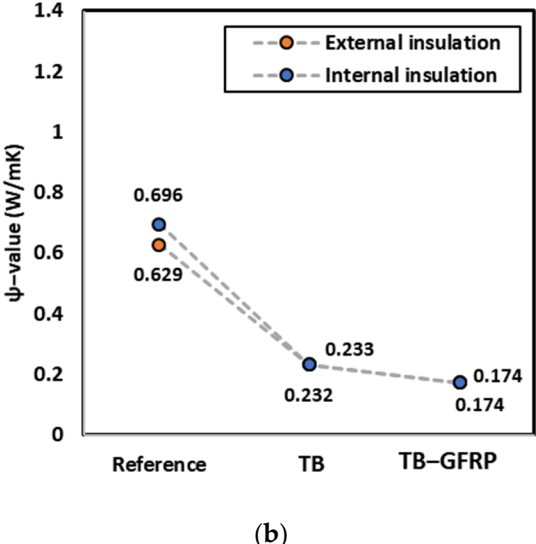

**(a)**　　　　　　　　　　　　　　　　　**(b)**

**Figure 5.** Simulation results under air heating condition: (**a**) Heat flow through the balcony–slab thermal bridge; (**b**) $\psi$-value.

With an internal insulation system, the reference case showed relatively larger heat flow (26.75 W) than that with external insulation system. The difference in the heat flow seems to be caused by the different heat flow path between external and internal insulation systems. The application of the TBs to the internal insulation system also showed the dramatic reduction in the heat flow. With TB (Case 5) and TB-GFRP (Case 6), the heat loss through the thermal bridge was reduced to 17.5 W and 16.32 W, respectively. The reduction rate was 34.6 for TB and 39.0% for TB-GFRP case, indicating that the TB-GFRP is more effective in mitigating the heat loss at thermal bridges.

Figure 5b shows the results of linear thermal transmittance ($\psi$-value) with thermal breaks. The $\psi$-values of reference thermal bridge (Case 1), the thermal bridge with TB (Case 2) and TB-GFRP (Case 3) were 0.629 W/mK, 0.232 W/mK, and 0.174 W/mK, respectively, when the external insulation system was applied. In case of the internal insulation system, $\psi$-values were 0.696 W/mK (Case 4), 0.233 W/mK (Case 5), and 0.174 W/mK (Case 6), respectively. It is shown that the $\psi$-value was reduced using TBs, regardless of the external or internal insulation systems. The reduction rate of $\psi$-value was 63.1–72.3% for the external insulation, and 66.5–75.0% in the internal insulation.

### 3.1.2. Radiant Floor Heating Condition

Table 6 shows the simulation results of heat transfer at the wall–slab junction under radiant floor heating condition. Compared to the results with air heating condition, the temperature inside the floor structure was considerably raised due to the heat supply by radiant floor heating system, which elevated the temperature difference between inside and outside balcony slab. As a result, the thermal bridge with radiant heating system showed much larger heat transfer than that with air heating system, as shown in Figure 6.

**Table 6.** Simulation results of heat transfer under radiant floor heating condition.

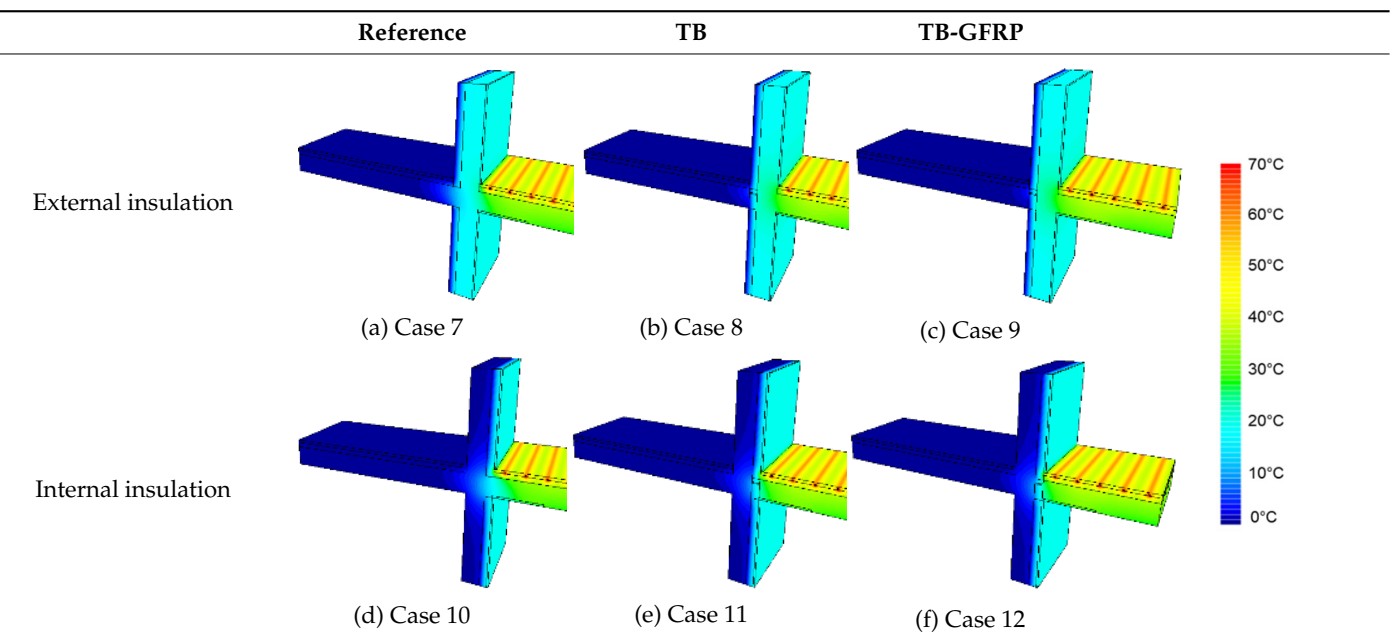

|  | Reference | TB | TB-GFRP |
|---|---|---|---|
| External insulation | (a) Case 7 | (b) Case 8 | (c) Case 9 |
| Internal insulation | (d) Case 10 | (e) Case 11 | (f) Case 12 |

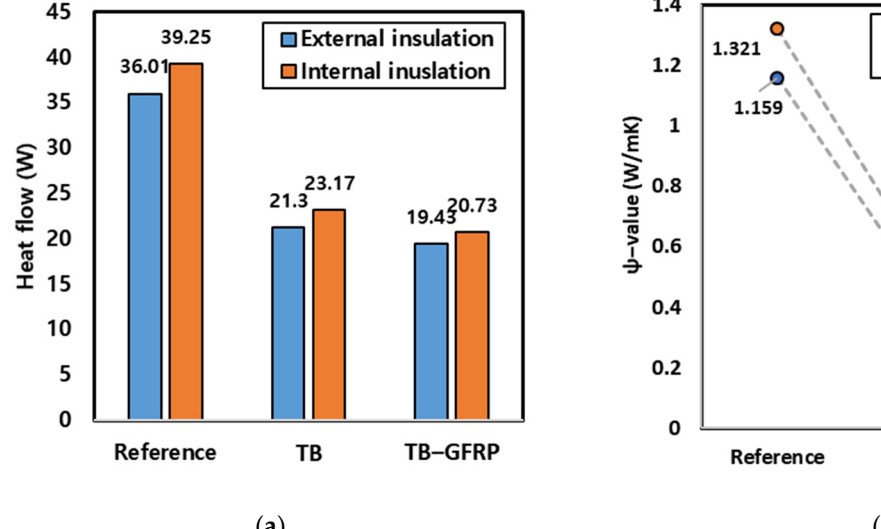
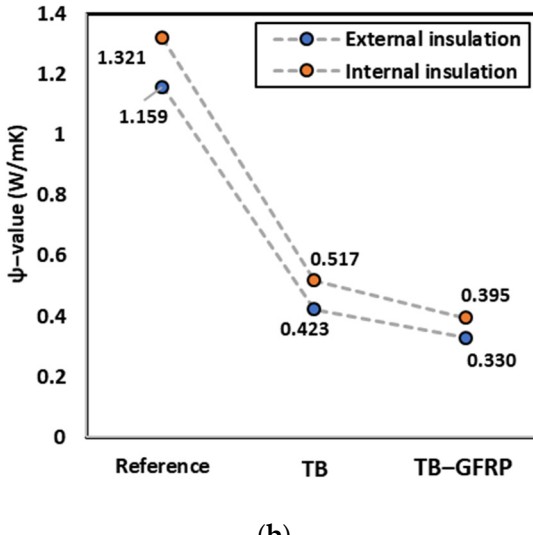

(**a**)　　　　　　　　　　　　　　　　　　　(**b**)

**Figure 6.** Simulation results under radiant floor heating condition: (**a**) Heat flow through the balcony–slab thermal bridge; (**b**) $\psi$-value.

Figure 6a shows that the thermal breaks are also effective in reducing heat flow through the balcony thermal bridge, even under radiant floor heating condition. In cases of external insulation system, the heat flow without thermal break was 36.01 W (Case 7), while the application of thermal breaks reduced the heat flow to 21.3 W (Case 8) and 19.43 W (Case 9). The reduction rate of the heat flow was 40.9% for TB and 46.0% for TB-GFRP case. With the internal insulation system, the heat flow of the reference case was 39.26 W (Case 10), while the application of thermal breaks reduced the heat flow down to 23.17 W (Case 11) and 20.73 W (Case 12), respectively. The reduction rate of heat flow was 41.0% for TB and 47.2% for TB-GFRP case.

The effect of thermal breaks on the heat flow reduction can be clearly evaluated using $\psi$-values. With the external insulation system, the $\psi$-value was 1.159 without thermal break (reference case), while it was reduced down to 0.423 W/mK ($-$63.5%) for TB and to 0.330 W/mK ($-$71.6%) for TB-GFRP. With the internal insulation system, the $\psi$-value of the reference case (1.321 W/mK) diminished to 0.517 W/mK ($-$60.9%) for TB and to 0.395 W/mK ($-$70.1%) for TB-GFRP.

Table 7 summarizes the result of linear thermal transmittance with the thermal breaks at the balcony slab. It is obviously shown that the thermal break has significant impact on the reduction in the heat loss through the balcony slab. The reduction effect was consistently found regardless of insulation method and heating system. Although the radiant floor heating system leads to the substantial increase in the $\psi$-value, thermal breaks effectively lessen the $\psi$-value, which would be efficient to improve the heating system performance. In addition, it should be noted that the TB-GFRP can reduce the $\psi$-value by more than 10%, compared to the TB. Thus, the thermal conductivity of reinforcing bars in thermal break block needs to be reduced in order to achieve better insulation performance at the thermal bridges.

**Table 7.** Summary of analysis results on the linear thermal transmittance.

| Insulation Method | Heating System | Reference | TB | TB-GFRP |
|---|---|---|---|---|
| Internal | Air heating | 0.696 W/mK | 0.233 W/mK (−66.5%) | 0.174 W/mK (−75.0%) |
| External | Air heating | 0.629 W/mK | 0.232 W/mK (−63.1%) | 0.174 W/mK (−72.3%) |
| Internal | Radiant floor heating | 1.321 W/mK | 0.517 W/mK (−60.9%) | 0.395 W/mK (−70.1%) |
| External | Radiant floor heating | 1.159 W/mK | 0.423 W/mK (−63.5%) | 0.330 W/mK (−71.5%) |

*3.2. Heating Energy Demand*

3.2.1. Air Heating Condition

Figure 7 shows the simulation result of annual heating energy demands with air heating system. In both cases of external and internal insulation, the heating energy demand reduced when the TB (Case 2, 5) and TB-GFRP (Case 3, 6) were applied. For external insulation systems, the reduction rates of Cases 2 and 3 were 5.6% and 6.5%, respectively, compared to the heating energy demand without thermal breaks (Case 1, 46.99 kWh/m$^2$). For internal insulation system, the heating energy demand without thermal breaks (Case 4) was 50.61 kWh/m$^2$. The application of TB (Case 5) and TB-GFRP (Case 6) reduced the heating energy demand by 6.1% and 7.0%, respectively.

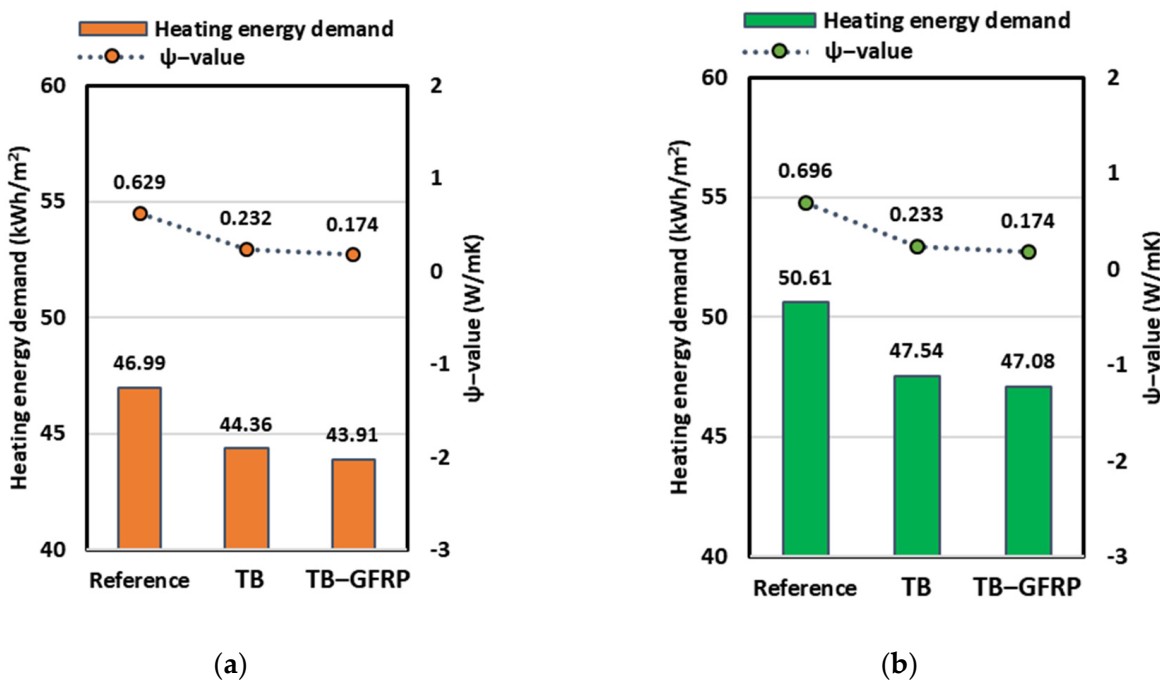

**(a)**                        **(b)**

**Figure 7.** Heating energy demand under air heating condition: (**a**) External insulation; (**b**) Internal insulation.

Peak heating loads for the analyzed cases were also evaluated, as shown in Figure 8. Similar to the heating energy demand results, the peak heating load also decreased with the application of thermal breaks. The reduction rate of peak heating load was up to 2.8% for the external insulation system and 3.1% for the internal insulation system. In both heating energy demand and peak heating load analysis, the TB-GFRP showed the biggest energy saving potentials at the balcony thermal bridge.

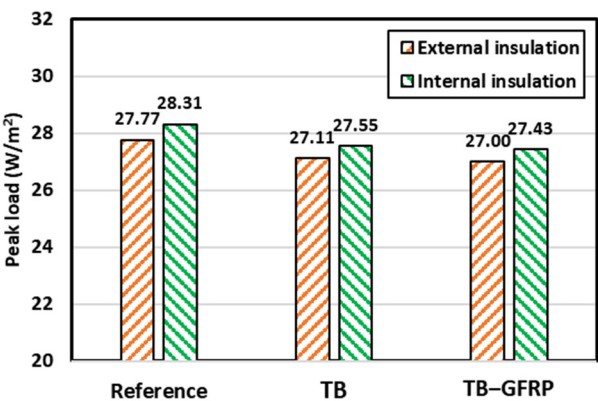

**Figure 8.** Peak heating load under air heating conditions.

### 3.2.2. Radiant Floor Heating Condition

Figure 9 shows the heating energy demand under the radiant floor heating condition. It is shown that the radiant floor heating condition resulted in more heating energy demand than the air heating condition, because the heated floor structure increased the heat flow through the balcony thermal bridge. Nonetheless, the heating energy demand reduced with application of the TB and TB-GFRP. The reduction rates were 8.0% with TB and 9.3% with TB-GFRP in case of internal insulation system. For external insulation system, the reduction rates were 7.7% with TB and 9.2% with TB-GFRP.

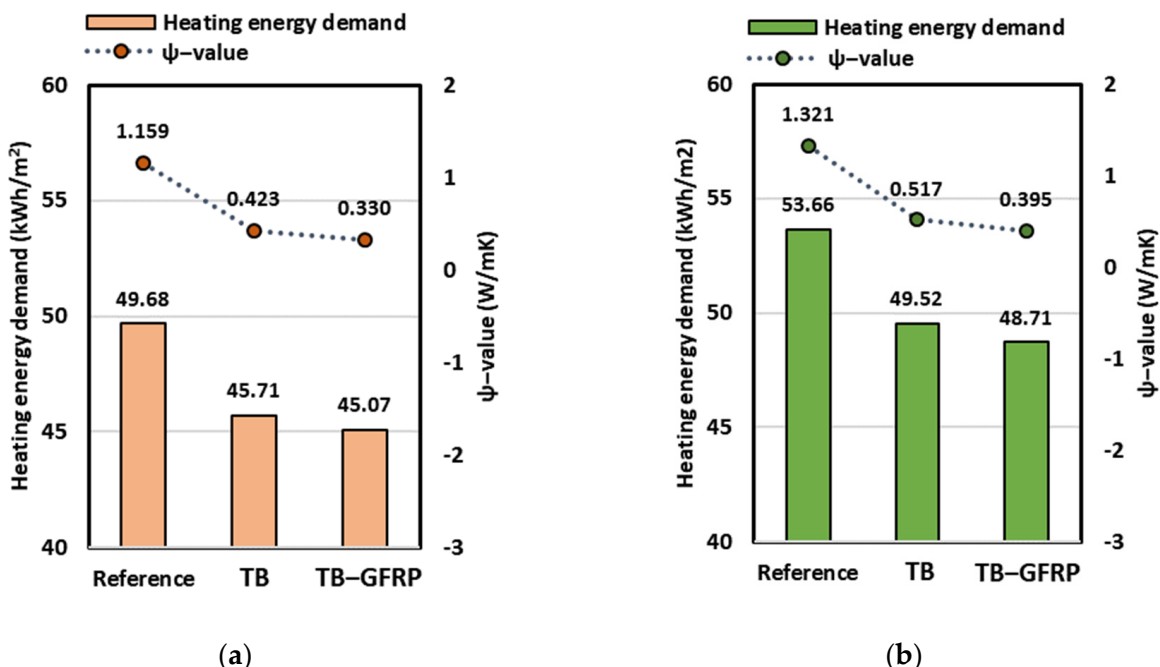

(**a**)
(**b**)

**Figure 9.** Heating energy demand under radiant floor heating condition: (**a**) External insulation; (**b**) Internal insulation.

Figure 10 shows the peak heating load under the radiant floor heating condition. Thermal breaks were effective in reducing the peak heating load as well as heating energy demand. The reduction rate of peak heating load was up to 4.0% for external insulation system and 4.2% for internal insulation system. It was found that the TB-GFRP can provide better energy saving performance under radiant floor heating condition, compared to the simple thermal break (TB).

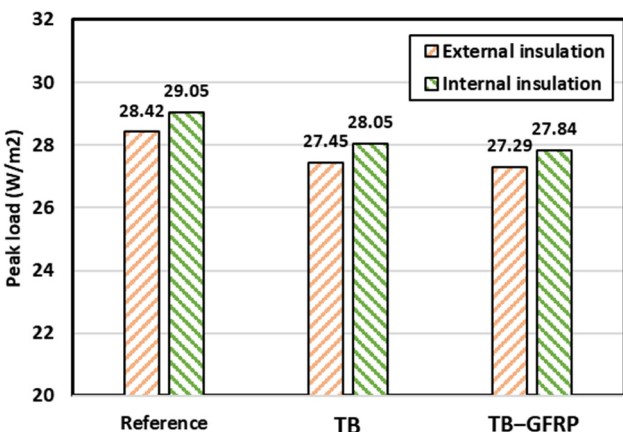

**Figure 10.** Peak heating load under radiant floor heating condition.

## 4. Discussion

Simulation results showed that the thermal break at balcony slabs is crucial to mitigate the heat loss through the balcony thermal bridge. It was also shown that the thermal break using reinforcing bars with the reduced thermal conductivity (i.e., GFRP) can provide additional energy saving effect. In addition, the thermal breaks have more impact on the energy saving for residential buildings equipped with radiant floor heating system, because the radiant floor heating system causes more heat transfer through the balcony slab, compared to conventional air heating systems.

Figure 11 summarizes the performance evaluation results of thermal breaks at balcony thermal bridge. It can be found that the heating energy demand has a linear relationship with the linear thermal transmittance at the balcony thermal bridge. Therefore, it is significant to reduce the linear thermal transmittance for the improvement of energy saving performance. Thermal breaks would be more significant for a residential building with more balconies, because the slope of the linear relationships between heating energy demand and linear thermal transmittance will be steeper.

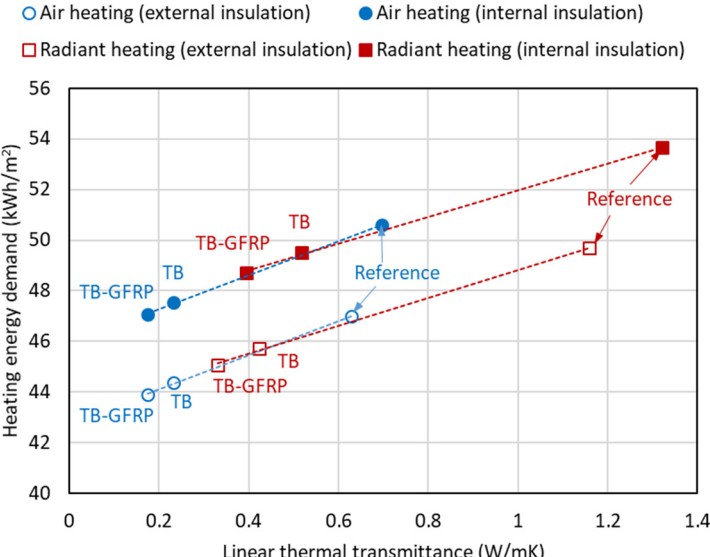

**Figure 11.** Linear thermal transmittance with heating energy demand.

It is also found that the radiant floor heating and internal insulation systems lead to relatively larger linear thermal transmittance and heating energy demand, compared to air heating and external insulation system, respectively. Thus, the air heating system with external insulation system can be considered as a more energy-efficient solution. If

it is inevitable to apply the radiant floor heating system with internal insulation system, thermal breaks should be applied to minimize the excessive heat loss thorough the balcony slabs. The application of the TB-GFRP rather than TB is recommended to improve the energy saving performance.

In this study, it was found that the thermal break is effective in reducing not only heating energy demand but also peak heating load. Thus, it would be possible to reduce the capacity of heating plants (e.g., boiler, heat exchanger) when the peak heating load is reduced by the application of thermal breaks. A further study is suggested to examine whether the heating capacity can be down-sized, considering the reduced peak heating load due to the thermal break.

This study was conducted to evaluate the impact of thermal breaks on heating energy demand. As the heating energy demand (or heating load) can be calculated without modeling HVAC system, this study implemented a so-called energy rate control approach [48] of TRNSYS simulation, wherein an ideal heat source supplies the required heat to maintain the set-point temperature. Nonetheless, a further study needs to be conducted with the HVAC systems modeled in order to investigate the energy consumption or economic feasibility of thermal breaks.

## 5. Conclusions

The objective of this study is to analyze the effect of a balcony slab thermal bridge on the heating energy demand of apartment buildings. To evaluate the heat loss caused by the thermal bridge, the linear thermal transmittance of the balcony slab thermal bridge was analyzed first. To reduce the heat loss at thermal bridges, two types of thermal breaks for balcony slabs (TB and TB-GFRP) were evaluated in terms of linear thermal transmittance and heating energy demand. The evaluation was conducted for different insulation methods (external and internal insulation) and heating conditions (air heating and radiant floor heating system).

The results showed that the TB and TB-GFRP reduce the linear thermal transmittance by 60.9–66.5% and 70.1–75.0%, respectively, depending on the insulation method and heating condition. The TB with low-conductivity reinforcing bars (TB-GFRP) showed the much reduction rate by 8.0–9.2% compared to the TB. It was clearly shown that the insulation performance of conventional TBs can be improved by replacing the steel reinforcing bars with the low-conductivity material such as GFRP. Annual simulation results showed that the TB and TB-GFRP can reduce the heating energy demand by 5.6–8.0% and 6.6–9.3%, respectively, depending on the insulation method and heating condition. It was also found that the TB-GFRP can save 0.9–1.5% more heating energy demand, compared to the TB. In particular, the impact of the TB-GFRP on the heating energy reduction was larger for a building with radiant floor heating system.

In conclusion, the investigated thermal break has potential for reducing heat loss through balcony thermal bridges. It is necessary to implement low-conductivity reinforcing bars (e.g., GFRP bars) for the thermal break in order to maximize the insulation effect of balcony thermal breaks. In particular, the thermal break needs to be applied to buildings with radiant floor heating system, because the heated slab structure can cause excessive heat loss at the thermal bridge.

**Author Contributions:** Conceptualization, K.-N.R. and G.-J.J.; methodology, X.Z. and K.-N.R.; soft water, X.Z. and K.-N.R.; validation, K.-N.R. and G.-J.J.; formal analysis, X.Z. and K.-N.R.; investigation, X.Z., K.-N.R. and G.-J.J.; resources, K.-N.R. and G.-J.J.; data curation, X.Z. and K.-N.R.; writing— original draft preparation, X.Z.; writing—review and editing, K.-N.R.; supervision, K.-N.R. All authors have read and agreed to the published version of the manuscript.

**Funding:** This work was supported by the Basic Science Research Program through the National Research Foundation of Korea (NRF-2019R1A2C1010515).

**Data Availability Statement:** Not applicable.

**Conflicts of Interest:** The authors declare no conflict of interest.

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
