# Peer review of "Performance Evaluation of Thermal Bridge Reduction Method for Balcony in Apartment Buildings"

_buildings, doi:10.3390/buildings12010063_

Round 1

Reviewer 1 Report

In this paper, different methods are evaluated in order to improve the balcony slab thermal bridge. The effects of thermal breaks elements on balcony thermal bridges are analyzed.

It constitutes a very interesting topic deserving in depth-study and publication in the scientific literature.  However, from this reviewer's point of view, this paper needs work to be accepted for the publication:

  • in the Introduction section there is an accurate bibliographic research, but this reviewer suggests to expand with more recent works;
  • add the references when the authors suggest values. I.e.: line 141 for the thermal conductivity of steel reinforcing bar used in the balcony slab; line 145 for the conductivities of the thermal break and stainless bars; etc. Please check in all the text;
  • Tables and Figures should be positioned after being cited in the text in order to make them easier to read. Please modify in all the manuscript (i.e. move Tab.1, Fig. 2, and Tab.2 on the next page; Fig. 3 after lines 214-220, and so on). Moreover, tables must not be divided into two pages (see table 2);
  • please explain TB in table 2;
  • merge the sub-sections 2.2.1, 2.2.2 in the single section 2.2 and rewrite with more explicitly and fluid way also resume when necessary (for example as concern the linear thermal transmittance theory);
  • line 236: which meteorological data were entered? What source was considered?
  • Were the HVAC systems modeled?
  • Section 2.2: please resume all input data of the simulation model in a table (after Fig. 4);
  • Pay attention to the layout of the Table 3 and Table 4;
  • Sections 3.1.1 and 3.1.2: please resume ψ-values obtained in the two cases in a table (resume the text with only discussion of the results in this table);
  • Lines 326 – 330: the same concepts were already expressed in the section above. Please modify or delete;
  • Sections 3.2.1 and 3.2.2: resume the results. The text is very redundant and in some parts it is boring. Try to summarize the results obtained by involving a fluid reading of the text;
  • Line 432: the authors declared: For this reason, it is necessary to analyze the economic feasibility by evaluating the energy consumption and return on investment (ROI) for a balcony slab in the analyzed cases. However, these analyses and the results not shown in the manuscript. Did the authors do it? please integrate this part;
  • Add in the conclusion section some numerical results.

Author Response

The authors are grateful for the valuable comments from the reviewer. Please see the attachment.

Reviewer 2 Report

The article presents results of investigation of impact of balcony to thermal load on buildings. Paper consider 3 cases of balcony construction and 2 cases of building insulation. Rasults are crealy shown on figures and tables. 

Some comments to authors:

  1. In line 51 page 2 is the internal wall temperature will rise when before is writen that loses are highier - isn't here some mistake?
  2. Page 4 Table 1 caption isn't changed from template
  3. In Table 1 in my opinion additionaly properties of standard balcony should be added
  4. Page 5 lines 168-172, I don't understand why thermal conductivity of both material was set to the same value, additionaly it not coresponding with the rest of data
  5. In the case of floor heating mean inside air temperature was obtained on the same value like in air heating? Because on figures 20C are only on pipes  so insiede building potentioaly we have lower thermal comfort.

Author Response

(The authors gave the same response as above.)

Reviewer 3 Report

This paper carries out a performance evaluation of thermal bridge reduction method for balcony slabs considering internal and external insulations, ISOKORB and TB-FGRP thermal break methods as well as underfloor heating and air heating systems. The paper is clear and well-written and structured. Please see below some minor amendments that could help to improve the quality of the paper:
L157: Please amend table 1 caption: "Table 1. This is a table. Tables should be placed in the main text near to the first time they are cited".
L163 Table 2: I suggest to replace Xs and Os with yes or no, to make it clearer.
L175: Percentages on the use of external or internal, if available, as well as under floor heating and air heating could help to depict how representative are these situations.
L392 Discussion: In this section you should only discuss the results. The overview of the research must be provided in the conclusions sector.
L47: 'Therefore' has been repeated. I suggest to use another word.

Author Response

(The authors gave the same response as above.)

Round 2

Reviewer 1 Report

-